# Effects of Functional Electrical Stimulation on Gait Characteristics in Healthy Individuals: A Systematic Review

**DOI:** 10.3390/s23218684

**Published:** 2023-10-24

**Authors:** Thomas Aout, Mickael Begon, Baptiste Jegou, Nicolas Peyrot, Teddy Caderby

**Affiliations:** 1Laboratoire IRISSE, EA4075, UFR des Sciences de l’Homme et de l’Environnement, Université de La Réunion, 97430 Le Tampon, France; baptiste.jegou40@gmail.com (B.J.); nicolas.peyrot@univ-lemans.fr (N.P.); teddy.caderby@univ-reunion.fr (T.C.); 2Laboratoire de Simulation et Modélisation du Mouvement, École de Kinésiologie et des Sciences de l’Activité Physique, Université de Montréal, Montreal, QC H3T 1J4, Canada; mickael.begon@umontreal.ca; 3Centre de Recherche du CHU Sainte-Justine, Université de Montréal, Montreal, QC H3C 3J7, Canada; 4Mouvement-Interactions-Performance (MIP), Le Mans Université, EA 4334, 72000 Le Mans, France

**Keywords:** electrical muscle stimulation, peripheral neuromodulation, walking, kinematics, kinetics, spatiotemporal, able-bodied

## Abstract

Background: This systematic review aimed to provide a comprehensive overview of the effects of functional electrical stimulation (FES) on gait characteristics in healthy individuals. Methods: Six electronic databases (PubMed, Embase, Epistemonikos, PEDro, COCHRANE Library, and Scopus) were searched for studies evaluating the effects of FES on spatiotemporal, kinematic, and kinetic gait parameters in healthy individuals. Two examiners evaluated the eligibility and quality of the included studies using the PEDro scale. Results: A total of 15 studies met the inclusion criteria. The findings from the literature reveal that FES can be used to modify lower-limb joint kinematics, i.e., to increase or reduce the range of motion of the hip, knee, and ankle joints. In addition, FES can be used to alter kinetics parameters, including ground reaction forces, center of pressure trajectory, or knee joint reaction force. As a consequence of these kinetics and kinematics changes, FES can lead to changes in spatiotemporal gait parameters, such as gait speed, step cadence, and stance duration. Conclusions: The findings of this review improve our understanding of the effects of FES on gait biomechanics in healthy individuals and highlight the potential of this technology as a training or assistive solution for improving gait performance in this population.

## 1. Introduction

Functional electrical stimulation (FES) is a peripheral neuromodulation technique that has been used in clinics for several years. This technique involves applying a low-intensity electrical current to neuromuscular tissues through either skin electrodes or directly implanted in the motor nerves. FES primarily aims to elicit involuntary muscle contractions in order to produce functional movements [1]. This widely recognized approach is crucial in restoring motor function and improving the quality of life for individuals with neuromuscular or neurological impairments [2,3]. Long-term FES use has been proven effective in functionally restoring and rehabilitating individuals with movement disorders, including stroke survivors and those with spinal cord injuries [4]. As an assistive technology, FES leads to enhanced functions, such as walking, maintaining a standing posture, and grasping, in these patients [5,6,7]. This stimulation method can also be employed as a short-term therapeutic strategy to restore unassisted mobility [2].

Since Liberson’s pioneering work in the 1960s [8], numerous applications of FES for gait assistance and restoration have been reported in the literature. In particular, FES has been widely used to address foot drop syndrome in patients with post-stroke hemiparesis [2,8], multiple sclerosis [9], and cerebral palsy [10]. By stimulating the fibular nerve or the tibialis anterior muscle of the weakened or paretic leg directly during a targeted period of the gait cycle, FES can increase ankle dorsiflexion angle during the swing phase and consequently toe clearance [8,9,10]. These FES-induced changes result in improved walking speed and a safer gait in the populations mentioned above [2,8,9]. Furthermore, FES has been applied to the plantar flexor muscles during terminal stance to increase the ankle plantar flexion angle at toe-off and leads to greater forward propulsion during walking in these patients [11]. Finally, FES has been applied to individuals with spinal cord injuries to regain standing posture and walking [4]. These outcomes were achieved by selectively and coordinately activating various lower limb muscles using surface electrodes or a neurostimulator implanted in the lumbar region [12]. All these findings indicate that FES is an effective solution for improving gait in people with neuromuscular impairments.

Beyond FES application in populations with neuromuscular disorders, FES has also been evaluated in healthy individuals to understand muscle function [13,14], validate musculoskeletal models [15], or improve gait performance [16]. Although several reviews have examined the influence of FES on gait parameters in people with neuromuscular impairments [17,18], to the best of our knowledge, no study has systematically reviewed the effects of FES on gait characteristics in healthy individuals. However, such information could provide valuable insights into the potential of FES in developing future assistive or training technologies. Therefore, this review aims to establish a systematic literature review about the effects of FES on the spatiotemporal, kinematic, and kinetic parameters of gait in healthy individuals.

## 2. Methods

This review followed the Preferred Reporting Items for Systematic Reviews and Meta-Analyses (PRISMA) 2020 statement. The protocol was registered on the Open Science Framework (https://osf.io/79gvq) accessed on 3 August 2023.

### 2.1. Search Strategy

Systematic searches were conducted using six scientific databases (PubMed, Embase, Epistemonikos, PEDro, COCHRANE Library, and Scopus) and cross-referencing. We reviewed articles written in English and French published up to 24 August 2023. The search terms were adjusted for each database (see Appendix A) and included: (“functional electrical stimulation”) AND (“healthy” OR “able-bodied” OR “normal” OR “non-disabled”) AND (“gait” OR “walking” OR “locomotion”). Two independent reviewers (TA and TC) independently screened articles according to the eligibility criteria.

### 2.2. Eligibility Criteria

Only peer-reviewed full-text articles meeting the following criteria were included in this review: (1) participants were healthy adults, with no restriction in terms of age, sex, or body mass; (2) the study assessed the acute or chronic effects of FES applied to one or more muscles during gait; and (3) the study tested at least one mechanical gait outcome (e.g., gait speed, spatiotemporal features, kinematics and kinetics parameters, etc.). Articles with the following features were excluded: (1) studies only assessing the combined effect of FES and another intervention (e.g., exoskeleton); (2) absence of a control condition (non-controlled design); and (3) case studies, case reports, conference papers, and book chapters. 

### 2.3. Study Selection Process 

The results identified by the search strategy were combined, and duplicates were removed. Two researchers (TA and TC) screened all papers independently. Articles were screened first by title and abstract, then full texts were checked based on the inclusion/exclusion criteria. Any disagreement during this selection process was resolved by discussion and mutual consent or by a third researcher (MFS).

### 2.4. Data Extraction

Data from all included studies were extracted by the first author (TA) and checked by the last author (TC). The characteristics of the participants (number, sex, and age), FES parameters (wave type, pulse width, frequency, intensity, localization, duration, and trigger), study protocol, outcome measures, and key findings were extracted from each study.

### 2.5. Methodological Quality Assessment

The methodological quality of the included studies was assessed independently by two researchers (TA and TC) using the Physiotherapy Evidence Database (PEDro) scale. The PEDro scale is a commonly used checklist consisting of 11 items rating the external and internal validity of studies. The total score of the PEDro scale corresponds to the number of “yes” responses for items 2 to 11, i.e., a total score expressed on 10 points. The first item is not included in the sum of the total score because it is the only item related to external validity [19]. Studies scoring ≥6 were considered “good” quality, those scoring 4 or 5 “fair” quality, and those scoring <4 were considered “poor” quality [20]. Any discrepancies between the two reviewers were resolved through discussion and mutual agreement or by a third researcher (MFS) providing a rating.

## 3. Results

### 3.1. Search Results

A total of 2583 records were initially identified from six databases. After removing 498 duplicates, 2085 studies underwent screening. Based on titles and abstracts, a blinded selection among these 2085 remaining studies was conducted. After analyzing the full texts, 15 studies met all criteria and were included in this systematic review. The flow diagram of the screening procedure is presented in Figure 1. Extracted data from the included studies are reported in Table 1. 

### 3.2. Methodological Quality of the Included Articles

Most studies [13,14,16,21,22,23,24,25,27,28,29] obtained a PEDro score of 6 or higher (n = 10), i.e., a good methodological quality (Table 2). The remaining five studies [15,22,24,26,31] scored 5, corresponding to a moderate methodological quality. The included studies are subject to common biases, especially in relation to items 6 and 7, which indicate that none of the examiners or researchers involved in these studies was blinded. Furthermore, 11 out of 15 studies did not clearly state the inclusion and exclusion criteria. Finally, seven studies did not randomize the experimental conditions.

### 3.3. Participants

A total of 215 healthy participants were included in these 15 studies. Fourteen studies examined the effects of FES on gait characteristics in healthy young individuals (186 participants; mean age: 25 years), whereas only one study investigated the effects of FES in the elderly (29 participants; mean age: 75 years) [16]. Two of these studies focused on the effects of FES during walking in healthy young subjects, comparing them to pathological populations such as post-stroke patients and individuals with chronic ankle instability [27,31]. In this systematic review, only data on healthy participants were considered and are summarized in Table 1.

### 3.4. Stimulation Parameters

*Targeted muscles*. The choice of stimulated muscles varied according to the objective of each study. Eight different muscles were targeted in the included studies (Table 3). The most frequently targeted muscle was the gastrocnemius [13,14,16,22,23,25,29,30], followed by the biceps femoris [13,15,16,23,25,26,30], tibialis anterior [13,16,23,25,27,30], rectus femoris [13,16,21,25,30], and soleus [13,14,22]. Additionally, isolated studies investigated the effects of FES on the gluteus medius [24], peroneus longus [31], and abductor hallucis [28].

*Wave type*. Regarding the type of current used, most of the experiments used a biphasic current [13,14,16,24,29,31]. Among them, three delivered a rectangular waveform stimulation [16,23,27] and two used an asymmetric waveform current [14,24].

*Pulse width*. In total, 14 studies mentioned pulse width. Several studies employed pulse widths of 250 μs [13,16,28], 300 μs [21,22], 350 μs [25,29], and 400 μs [24,27]. Only one study used a pulse width of 120 μs [15], whereas another used 200 μs [31]. Notably, one of the fourteen studies mentioned implemented adjustments to the pulse widths based on participant characteristics [14,29]. Moreover, two additional studies employed variable pulse widths ranging from 0 to 250 μs [23] and from 250 to 500 μs [30].

*Stimulation frequency*. Thirteen studies mentioned the stimulation frequency. Among these studies, nine used a stimulation frequency of 40 Hz [13,14,15,16,25,26,27,29,31]. Two studies used a stimulation frequency of 33 Hz [15,22], whereas two others employed frequencies of 45 Hz [24] and 65 Hz [23].

*Stimulation intensity*. Twelve studies indicated using a stimulation intensity above the motor threshold [13,15,24,25,26,29,31] and/or below the pain threshold [13,14,16,24,25,26,28,29,31]. Four of these studies determined the stimulation intensity based on a desired mechanical output. Specifically, two studies determined the appropriate stimulation intensity based on the knee flexion angle [15,26]. In studies [24,27], FES intensity was determined upon reaching a specific angle at the ankle (neutral position) or hip (30–45° abduction), respectively. Among the 10 studies that evaluated stimulation intensity relative to the pain threshold, one study employed an intensity equivalent to a subjective value of 2 on a 10-point visual analog pain scale [21] and another an intensity corresponding to 2/3 of the maximum tolerance threshold [29].

*Stimulation timing*. The majority of included studies provided explicit details about the timing of muscle stimulation within the FES protocol. These details are displayed in Table 1 and Table 3. In seven studies, electrical muscle stimulation was applied between temporal events of the gait cycle [14,21,25,26,27,28,30,31], described hereafter according to the targeted muscle: abductor hallucis muscle from foot-flat to heel-off (mid-stance phase) [28], biceps femoris from heel-strike to toe-off (stance phase) [26] or from toe-off to heel-strike (swing phase) [25,30], rectus femoris from terminal swing to subsequent heel-off [25,30], peroneus longus muscle from 0% to 80% of the stance phase [31], tibialis anterior from heel-off to heel-strike [27] or from toe-off to heel-strike (swing phase) [25,30], gastrocnemius lateralis and soleus muscles (separately) from heel-strike to foot-flat (loading response) [14], from foot-flat to toe-off [14], from heel-off to toe-off (pre-swing phase) [14], or from heel-strike to toe-off (stance phase, only for the gastrocnemius lateralis) [25,30]. Alternatively, five studies applied the stimulation at a specific instant of the gait cycle for a fixed duration [13,21,24,25,29,30]: one study stimulated the gastrocnemius medialis muscle for a duration of 100 ms at eight different instants of the stance phase and four distinct instants of the swing phase [29]; two other studies separately stimulated the soleus and gastrocnemius medialis muscles for a 90 ms duration either at 20% (mid stance) or at 30% (terminal stance) of the gait cycle [13,22]; and another study stimulated the rectus femoris muscle for 90 ms either at 50% (pre-swing) or 60% (early-swing) of the gait cycle [21]. Furthermore, studies stimulated the tibialis anterior [16,27], biceps femoris [16], gastrocnemius medialis [16], and quadriceps [16] muscles according to the activation sequence of these muscles during the gait cycle detected by electromyography [16]. Another study mentioned stimulating these same four muscles, based on the signal of a knee electrogoniometer, in a such way that the timing activation sequence corresponded to the observed muscle activation patterns during normal gait [23]. However, no information was provided on the timings used to trigger stimulations from the electrogoniometer. Finally, two other studies stimulated the biceps femoris [15] and gluteus medius [24] during the stance phase of gait. However, they did not provide further details or specific timing information regarding the exact moments of the stimulations within this phase. 

*Triggering method*. Each of the analyzed studies employed a different method to trigger stimulation during walking. Most of these studies triggered stimulation solely using foot-sensitive resistors [14,27,31] or combining this trigger with an inertial measurement unit [25,30]. Three studies used a manual trigger (hand switch) to initiate stimulation during walking [15,26,28], whereas three other studies triggered stimulation based on ground reaction force signals from force plates [13,21,22]. In parallel, three studies respectively triggered stimulation based on the angular velocity signal from an inertial measurement unit [29], on an electromyography signal [16], or on a joint angle using a goniometer signal [23]. Finally, only one study did not specify the type of trigger used [24].

### 3.5. Spatiotemporal Parameters

Out of the 15 studies examined, only 6 addressed the effects of stimulation on the spatiotemporal parameters of gait [15,16,23,26,28,29]. The findings from these studies are summarized in Table 1 and Table 3. 

*Walking speed*. Five studies reported the effects of FES on walking speed. Only Park et al. [16] found an 11% increase in walking speed by applying FES to the biceps femoris, rectus femoris, tibialis anterior, and gastrocnemius medialis muscles. In contrast, Azmi et al. [26] revealed a 7% decrease in walking speed when stimulating the biceps femoris long head muscle during the stance phase. Three other studies did not observe any significant effects of FES on walking speed [15,23,28]. The latter studies applied stimulation to the biceps femoris during the stance phase [15], the abductor hallucis during the mid-stance phase [28], and to the tibialis anterior, biceps femoris, gastrocnemius medialis, and quadriceps muscles according to their activation sequence during normal gait [23]. 

*Step frequency, length, and width*. Regarding step frequency, one study demonstrated a 13.5% increase in this parameter by applying FES to the biceps femoris, rectus femoris, tibialis anterior, and gastrocnemius medialis muscles during overground walking [16]. In contrast, another study observed no effect on this parameter when stimulating the same muscles during treadmill walking [23]. Neither of the latter two studies found significant effects of FES on step width, step length, or stride length [16,23].

*Stance and stride duration*. Two studies reported an increase in stance time. In the first study, this parameter was increased by stimulating the biceps femoris, quadriceps, tibialis anterior, and gastrocnemius medialis muscles according to their activation sequence during normal gait [23]. The second study observed a longer stride duration when stimulation of the gastrocnemius medialis muscle was triggered around foot contact with the ground [29]. However, a shorter stride duration was noted when stimulation was applied around the propulsive phase [29]. Finally, no significant effect was found with stimulation of the abductor hallucis from mid-stance to pre-swing [28]. 

### 3.6. Gait Kinematics

Several studies investigated the effects of FES on gait kinematics (Table 3). Eleven studies have examined the effects of FES on joint and segment angles during overground walking [13,14,24,26,28] or on a treadmill [21,25,27,30,31].

*Ankle kinematics*. Regarding ankle joint, three studies showed an increase in peak plantar flexion angle during walking with three different stimulation configurations. The peak plantar flexion angle during walking was examined with three different stimulation configurations. In the first configuration, the ankle plantar flexion angle increased when stimulating the soleus from foot-flat to toe-off, whereas no significant results were obtained when the stimulation was triggered from initial contact to foot-flat or from heel-off to toe-off [14]. The second configuration resulted in a 1° increase in ankle plantar flexion angle by stimulating the soleus during the mid-stance phase [13]. Finally, the third configuration involved the stimulation of the gastrocnemius lateralis during the stance phase, in combination with the stimulation of the tibialis anterior, biceps femoris, and rectus femoris throughout the gait cycle [25]. Chen et al. [27] found that stimulation of the tibialis anterior from heel-off to heel-strike resulted in a 4.3° decrease in the plantar flexion angle at toe-off during walking at 1.2 m/s (but not at 0.3, 0.6, or 0.9 m/s), whereas there were no significant changes in this parameter when the stimulation was applied to this muscle during its activation sequence determined from electromyography. In parallel, four configurations demonstrated an increased ankle dorsiflexion angle during walking trials conducted on the ground [13,14] or on a treadmill [25,27]. Stewart et al. [14] reported an increased ankle dorsiflexion angle by stimulating the gastrocnemius lateralis from foot-flat to toe-off, whereas Lenhart et al. [13] obtained a 0.7° increase in ankle dorsiflexion angle by triggering gastrocnemius medialis stimulation during mid-stance. Moreover, Chen et al. [27], by using FES triggered by a speed-adaptive algorithm, achieved a 2.8° increase in ankle dorsiflexion angle during the swing phase at a speed of 1.2 m/s (but not at 0.3, 0.6, or 0.9 m/s) compared with trials without stimulation. Lastly, Meng et al. [25] increased ankle dorsiflexion by stimulating the tibialis anterior in combination with the stimulation of other muscles during the gait cycle, including the gastrocnemius lateralis, biceps femoris, and rectus femoris. Additionally, only Gottlieb et al. [31] mentioned the effects of FES on ankle eversion. They found that a single session of walking training with a stimulation of the peroneus longus induced a 2° increase in ankle eversion during early stance (0–9% of the stance phase) and a 1° increase in ankle eversion during the late stance phase (82–89% of the stance phase) in the healthy control group. Lastly, only the study conducted by Dong et al. [30] showed a 36.5% (i.e., about 9.1°) increase in the ankle range of motion (ROM) in the sagittal plane; this was in the condition using an adaptative and reflexive FES controller compared with a condition without FES.

*Knee kinematics*. Regarding the knee joint, six studies reported results related to knee flexion angle in the sagittal plane [13,14,21,25,27,30]. The knee flexion angle was successfully increased by stimulating the gastrocnemius lateralis in isolation between the instants of foot-flat and toe-off [14] or in combination with the stimulation of the tibialis anterior, biceps femoris, and rectus femoris during the pre-swing phase [25]. Two other studies indicated a 17.6% increase (i.e., about 9.4°) in the peak knee flexion angle by stimulating the rectus femoris, biceps femoris, gastrocnemius lateralis, and tibialis anterior on both legs [30] and a 3.2° increase in motion on average by stimulating the gastrocnemius medialis at 20% of the gait cycle [13] or a 1.9° increase in motion on average by stimulating the same muscle at 30% of the gait cycle [13]. In contrast, another study found a 7.5° decrease in knee flexion angle during walking trials by stimulating the rectus femoris on the right leg during the pre-swing phase [21] and a 1.7° decrease in knee flexion angle at the early-swing phase [21]. Chen et al. [27] reported that the stimulation of the tibialis anterior between heel-off to heel-strike had no effect on knee flexion angle, but the stimulation of this muscle applied during its activation sequence determined by electromyography resulted in a 1.1° increase during walking at 0.9 m/s (but not at 0.3, 0.6, or 1.2 m/s). Regarding knee extension angle, two studies reported a 1° increase in knee extension angle by stimulating the soleus at the mid-stance [14], a 0.6° increase by stimulating the soleus at the terminal stance [14], or a 3.4° increase by stimulating the rectus femoris and biceps femoris during the terminal swing in combination with the stimulation of the gastrocnemius lateralis and tibialis anterior through the gait cycle [30]. Only Meng et al. [25] showed decreased knee extension angle when the gastrocnemius lateralis was stimulated during the loading response in combination with the stimulation of the biceps femoris, rectus femoris and tibialis anterior through the gait cycle. Lastly, only the study conducted by Dong et al. [30] showed a 21.6% (i.e., about 12.9°) increase in the knee ROM in the sagittal plane; this was in the condition using an adaptative and reflexive FES controller compared with a condition without FES.

*Hip kinematics*. Some studies investigated the effects of FES on hip kinematics. Three studies reported a 1.5° increase in the peak hip flexion angle through the stimulation of the gastrocnemius medialis during the mid-stance phase [13] and a 17% (i.e., about 4.8°) increase in the hip flexion angle by stimulating the rectus femoris and the biceps femoris during the terminal swing in combination with the stimulation of the gastrocnemius lateralis and tibialis anterior trough the gait cycle [25,30]. In contrast, a study showed a decrease in the peak hip flexion angle by stimulating the rectus femoris during the pre-swing phase and during the early swing phase [21]. Regarding hip extension angle, two studies reported an increase in peak extension angle when stimulating the biceps femoris and the rectus femoris during the terminal swing in combination with the stimulation of the gastrocnemius lateralis and the tibialis anterior through the gait cycle [25,30], along with a decrease in hip extension angle during the stance phase [25]. Finally, these last two studies showed an increase in the hip ROM in the sagittal plane [25,30]. One of them specified a 20% (i.e., about 7.6°) increase in the hip ROM in the condition using an adaptative and reflexive FES controller compared with a condition without FES [30].

*Pelvic kinematics*. Only two studies reported pelvic kinematics [13,24]. The first one showed a more posteriorly pelvis tilt during the mid-stance phase, occurring 200 ms after the onset of stimulation of the gastrocnemius medialis [13]. Moreover, the same study showed a 0.4° increase in anteriorly pelvis tilt by stimulating the soleus at mid-stance [13] and a 0.3° increase in anteriorly pelvis tilt by stimulating the soleus during the terminal stance [13]. The second study demonstrated a 46% reduction in the pelvis drop towards the swing leg (in the frontal plane), with a stimulation of the gluteus medius during the terminal stance [24].

*Foot kinematics*. Among the studies reviewed, only Okamura et al. [28] looked at the effects of FES on foot kinematics during walking. In the FES group, where the abductor hallucis was stimulated from the mid-stance phase to the pre-swing, the timing of the minimum navicular height was significantly later than in the group without FES. Additionally, this study revealed a 17% reduction in the forefoot abduction angle (in the transverse plane) relative to the rear foot in the FES group. However, no significant differences were observed in the changes of forefoot dorsiflexion (in the sagittal plane) and eversion (in the frontal plane) angles. 

### 3.7. Gait Kinetics

Four studies examined the effects of FES on gait kinetics during trials conducted exclusively on level ground [22,24,26,28]. The findings from these studies were summarized and displayed in Table 1 and Table 3. 

*Ground reaction force*. Among the included studies, three of them investigated the effects of FES on ground reaction forces [22,24,28]. Okamura et al. [28] observed a decrease in the peak vertical ground reaction force (second peak) by stimulating the abductor hallucis from the mid-stance to the pre-swing. Francis et al. [22] also showed an increase in the vertical ground reaction force by stimulating the soleus at 20% of the gait cycle. Additionally, the same study indicated an increased anteroposterior ground reaction force by delivering stimulation to the gastrocnemius medialis at 30% of the gait cycle [22]. Rane and Bull [24] observed that stimulating the gluteus medius during the terminal stance reduced the impulse of mediolateral ground reaction force by 18% during the stance phase and decreased the vertical ground reaction force impulse, whereas the same stimulation timing of the gluteus medius increased the impulse of the anteroposterior ground reaction force [24]. Conversely, a study stimulating the soleus at 20% and at 30% of the gait cycle showed a decrease in anteroposterior ground reaction force for both timings [22]. 

Two studies examined the forces generated at the knee during FES-assisted walking [24,26]. The first study showed that stimulation applied to the biceps femoris long head during the early stance reduced anterior tibiofemoral shear forces [26]. Furthermore, this stimulation contributed to a 144% increase in the compressive force applied to the lateral condyle of the knee and a 63% decrease in the peak internal tibial rotation torque [26]. In the second study, the researchers observed a 4.2% decrease on average in the impulse of the medial knee joint reaction force, with decreases of 6.5% in the magnitude of the mid-stance impulse and 3.9% in the terminal stance impulse by stimulating the gluteus medius of the right leg prior to the right foot strike [24]. In the same study, mean reductions in peak force with FES were 13.8% for the first peak and 18.4% for the second peak of the medial knee joint reaction force [24].

*Center of pressure*. Only the study conducted by Francis et al. [22] examined the displacement of the center of pressure (CoP) during FES-assisted walking. This study showed an anterior displacement of the CoP when the soleus was stimulated during the mid-stance phase. Similar results were obtained with stimulation of the gastrocnemius medialis during the mid-stance phase [22].

## 4. Discussion

The aim of this article was to perform a systematic review of the studies investigating effects of FES on gait characteristics in healthy individuals. Fifteen studies were included. Overall, the findings of this review indicate that the application of FES to one or more muscles when walking changes various gait parameters, including spatiotemporal parameters, joint and body segment kinematics and kinetics, and ground reaction forces. However, the effects of FES on these parameters varied depending on the stimulated muscles, the timing, and the stimulation parameters.

### 4.1. FES Parameters during Gait in Healthy Individuals

All the studies included in this review focused on the effects of FES delivered to muscles of the lower limbs, namely the gastrocnemius, biceps femoris, rectus femoris, tibialis anterior, soleus, abductor hallucis, gluteus medius, and peroneus longus. These 15 studies administered electrical muscle stimulation during the phases of physiological activation detected with different devices during normal walking [32]. Depending on the studies, the stimulation partially intervened or covered the entire duration of these activation sequences to increase the level of muscle activation and enhance their agonist or antagonist action during movement. 

The effects of FES on muscle activation depend on various stimulation parameters, including the intensity, frequency, duration, type of current, and pulse width. All the analyzed studies used a stimulation intensity ranging between the motor threshold and the pain or discomfort threshold. This choice ensures that the stimulation produces a motor effect without causing discomfort to the participant. Stimulation intensity, which corresponds to the current amplitude administered during stimulation, modulates the motor units’ recruitment and influences the level of muscle fatigue [33]. Thus, a high intensity would increase the level of force and muscle fatigue generated, whereas low-intensity stimulation would result in the opposite effect [33]. Moreover, recent research suggests that high-intensity stimulation not only induces an increase in muscle fiber contraction force but can also lead to antidromic transmission [34]. This particular type of neural transmission originates from the nerve fiber and travels towards the spinal cord, resulting in the inhibition of sensory and motor impulses originating from the motor neuron pool. Consequently, excessive stimulation intensity reduces the overall activation of the central nervous system [34]. On the other hand, although very low-intensity stimulation promotes orthodromic transmission from the motor neuron pool to the target muscles, it may elicit no motor response [34]. Therefore, it is advisable to determine an optimal stimulation intensity that maximizes the effects induced by FES while minimizing the discomfort the participants perceive during its application.

Additionally, the type of current and stimulation frequency are essential parameters that influence the effects produced by FES due to their impact on motor unit recruitment patterns. Among the included studies, nine delivered muscle electrical stimulation at a frequency of 40 Hz, with a biphasic current in six of them. The choice of these parameters appears relevant because the use of a biphasic wave frequency pattern between 20 Hz and 50 Hz is perceived as clinically more effective and comfortable [35]. Furthermore, this frequency choice seems appropriate as its value corresponds to the patterns used to minimize muscle fatigue occurrence [36,37]. Dreibati et al. [35] highlight that a stimulation frequency below 50 Hz mainly recruits slow-twitch-type motor units, characterized by slow contraction and increased fatigue resistance. Conversely, higher frequencies lead to greater recruitment of fast-twitch motor units (type IIa and IIb), which contract rapidly but fatigue more quickly [38]. These high frequencies can be efficient when fast contractions are required to meet the temporal constraints imposed by the gait cycle. For instance, when a muscle needs to generate force within a very short timeframe, employing high frequencies can enhance the rate of force development [39]. Consequently, the stimulation frequency is a crucial parameter that significantly impacts the onset of fatigue, the level of force produced, and the rate of force development. A stimulation frequency aligned with the physiological discharge rates of the motor unit seems to be a good compromise to limit muscle fatigue onset while preserving force production [40,41].

Regarding pulse width, most studies included in the analysis employed pulse widths spanning from 200 to 400 μs [13,15,16,21,22,24,28,29,30,31]. Additionally, two studies utilized a variable pulse width configuration [23,30], whereas one study customized these values based on participant characteristics [14]. This stimulation parameter is crucial as its value directly influences the recruitment of muscle fascicles per pulse [42]. Grill et al. [42] observed that short pulse widths (10 μs and 50 μs) lead to a reduction in the recruitment of muscle fascicles. Hence, increasing the pulse width could potentially recruit adjacent fibers as fatigue sets in [42]. Investigations targeting soleus stimulation with varying pulse widths (50, 200, 500, and 1000 μs) highlighted that higher pulse widths resulted in more pronounced contractions during plantar flexion [43]. Furthermore, longer pulse widths enable deeper tissue penetration of the stimulation and should be considered when applying FES to deep muscles [44].

### 4.2. Effects of FES on Gait Parameters in Healthy Individuals

Several studies have investigated the effects of FES on joint and segmental kinematics during walking. Overall, the studies reported that FES produced different results depending on the targeted muscle and the stimulation timing. FES proves to be an effective means for enhancing ankle plantar flexion and dorsiflexion angles during the stance [14], mid-stance [13], pre-swing [25], and swing phases [25,27]. Additionally, this technique is able to increase ankle eversion during the early and late stance [31]. Regarding knee angles, FES can increase the knee flexion angle during the stance [14], mid-stance [13], or swing phases [25] and increase the knee extension angle during the stance phase [14]. Lastly, this stimulation method increased the hip flexion and extension angles [25,30].

Conversely, FES can also be used to reduce the knee flexion angle during the pre-swing and early swing phases [21], decrease knee extension angles during the stance phase [25], restrict pelvic drop during the swing phase [24], reduce longitudinal arch deformity during the stance phase [28], or reduce hip flexion angle only during the stance phase [25]. Although it has been shown that FES is able to increase the agonist or antagonist action of muscles in the lower limbs, it is interesting to note that the action of the same muscle can be modified depending on the timing of stimulation. Indeed, opposite actions have been observed for the knee flexion angle when stimulating the rectus femoris at the mid-swing (increase in knee flexion angle) [30] compared with stimulation triggered at the pre-swing [21] or early swing (decrease in knee flexion angle) [21]. Similarly, changes have been found when stimulating the gastrocnemius medialis [13] at mid-stance (increased knee flexion angle) compared with the terminal stance (decreased knee flexion angle) [13]. 

As for the amplitudes of these changes, the clinical application of FES remains relevant even in light of the subtle alterations it introduces into the joint kinematics of healthy individuals. Several studies investigating elderly fall-related issues have highlighted that minimal changes in joint ranges are observable between fallers and non-fallers. For instance, Chiba et al. [45], examining toe clearance during the swing phase, noted a 3 mm discrepancy between the two groups. Similarly, Menz et al. [46] found a 3.5° variance in ankle flexibility between fallers and non-fallers. Furthermore, Kerringan et al. [47] identified a 3° distinction in hip extension between fallers and non-fallers during gait trials conducted at preferred speed. 

In summary, FES can amplify joint angles when applied to agonist muscles and reduce them when delivered to antagonist muscles. Moreover, opposite effects may be obtained according to the stimulation timing for a same stimulated muscle. Finally, modifications induced by FES, even minimal ones, have the potential to substantially enhance gait patterns in healthy individuals.

In addition to gait kinematics, various studies included in this review reported that FES was an effective means for changing gait kinetics. Among these studies, two reported changes in ground reaction forces and CoP by stimulating various muscles belonging to the lower limbs at different timings of the gait cycle. Regarding ground reaction forces, two studies found an increase in second peak vertical force by stimulating the abductor hallucis from mid-stance to pre-swing [28], the soleus at 20% of the gait cycle [22], and the gastrocnemius medialis at 20% and 30% of the gait cycle [22]. Moreover, contrasting results were obtained concerning the anteroposterior component of the ground reaction force by stimulating the soleus and the gastrocnemius medialis during the stance phase. Francis et al. [22] showed an increase in this anteroposterior force 150 ms after the onset of gastrocnemius medialis stimulation at 30% of the gait cycle. Conversely, a decrease in the anteroposterior component was obtained 100 ms after the soleus stimulation onset at 30% of the gait cycle and 50 ms after the soleus stimulation onset at 20% [22]. These results are accompanied by a CoP anterior displacement 100 ms after the soleus and gastrocnemius medialis stimulation onset at 20% of the gait cycle. Hence, the stimulation timing seems to influence the contribution of these plantar flexor muscles in the forward propulsion [22]. These findings align with earlier studies conducted by Kimmel et al. [48] and Neptune et al. [49], which indicated that the soleus can function as a braking mechanism during its early activation (mid-stance) but can contribute to forward propulsion when activated later in the gait cycle (terminal stance and pre-swing).

Concerning the ground reaction force impulse, only the study conducted by Rane and Bull [24] indicates an increase in the anteroposterior component when the gluteus medius is at its maximum stimulation intensity during the stance phase [24]. Furthermore, the same stimulation reduced the vertical ground reaction force impulse and the mediolateral ground reaction force impulse by 18% [24]. According to this study, the stimulation of the gluteus medius seems to lead to an increase in forward braking and a lateral shift of the body’s center of mass toward the stance leg [24]. 

Regarding the effects of FES on muscle and joint forces, three studies showed that using FES during the stance phase modifies the force impulses of the biceps femoris long head and the gluteus medius [15,24]. Moreover, increased activation of the gluteus medius and biceps femoris long head has been found to alter the forces applied to the knee during walking [24,26]. Stimulating the gluteus medius during the stance phase resulted in a 12.5% overall reduction in the medial knee joint reaction force impulse during mid-stance and terminal stance. Additionally, there was an average reduction of 13.8% for the first peak and 18.4% for the second peak [24]. Concerning the biceps femoris long head, stimulation of this muscle led to a reduction in the anterior tibiofemoral shear forces, 144% increased total compressive forces applied to the lateral knee condyle, and a 63% decrease in the peak of internal tibial rotation torque [26]. Therefore, the findings suggest that FES can effectively modify muscle and joint forces during walking in healthy individuals, potentially offering benefits for gait biomechanics and joint loading.

Several studies have investigated the effects of FES on spatiotemporal parameters during overground [15,16,26,28] or treadmill walking [23,29]. The findings regarding walking speed vary across the studies. Park et al. observed an 11% increase in walking speed in an older population (mean age: 75) when stimulating the rectus femoris, the biceps femoris, the tibialis anterior, and the gastrocnemius lateralis at different moments of the stance and swing phases, based on the activation sequences of these muscles determined from electromyography [16]. However, Azmi et al. reported a 7% decrease in walking speed in a younger population (mean age: 26 years) when stimulating the biceps femoris long head during the stance phase [26]. Moreover, three studies conducted with young individuals found no effect of FES on walking speed during overground walking [15,28] or treadmill walking [23]. Additionally, two studies reported an increased stance time when agonist and antagonist muscles of the leg and thigh were stimulated [23,29]. These results were associated with a slowing of the stride during foot–ground contact and an acceleration of the stride around the propulsive phase during treadmill walking [29]. Only one study showed a 13.5% increase in step frequency when the muscles of the leg and thigh were stimulated at different moments of the stance and the swing phases during overground walking trials [16]. Lastly, the studies mentioned earlier found no significant effect of FES on step width, step frequency, or stride length during overground and treadmill walking [16,23,28]. In conclusion, these findings suggest that FES can modify spatiotemporal parameters during gait. However, the effects can vary based on factors such as the population type (older or young individuals), the stimulated muscles, and their stimulation timing.

### 4.3. Applications, Perspectives, and Limitations

Overall, the results suggest that FES is an effective tool for modifying gait characteristics in healthy individuals. Its use in the healthy population could have many applications. Firstly, FES may enhance the understanding of muscle roles during human gait, thereby facilitating the development and validation of more sophisticated musculoskeletal models [15]. Furthermore, this technology can potentially contribute to developing solutions aimed at reducing or altering mechanical stress on human biological tissues, such as bones, muscles, tendons, or cartilage [24,26]. Finally, FES could also be exploited as an assistive or training solution to improve the walking performance in both healthy individuals and those with mobility impairments, including the elderly [16]. In particular, it would be relevant to investigate the effects of FES on balance and energetics during walking in the elderly population. This is crucial considering that the increased risk of falls and heightened energy expenditure during walking in this demographic appear to be primarily associated with muscular factors [50,51,52,53].

Concurrently, recent studies involving individuals with neurodegenerative pathologies have revealed recurrent temporal and kinematic alterations during gait compared with healthy subjects [54,55]. In this context, a deeper understanding of the effects of FES on a healthy population could serve as a robust foundation for developing biotechnological solutions aimed at mitigating the adverse effects observed during the gait of individuals with neurodegenerative diseases.

It is important to acknowledge the limitations of this review. The studies included in this review had varying objectives and methodologies. Furthermore, there was variation in the FES devices used and their configurations across the studies. Consequently, the substantial variability observed between studies can make it difficult to interpret and compare results. Nevertheless, overall, the results of these studies demonstrate that electrical stimulation can be used to modify the kinematic, kinetic, and spatiotemporal parameters of walking in healthy individuals. Moreover, a limitation of this review is the absence of an examination of the effects of FES on muscle activity (EMG) during walking. Considering the diverse body of research on this topic, it is not feasible to comprehensively address all the findings from these studies within the scope of this systematic review. Nevertheless, due to the extensive nature of this subject, it warrants further attention in future research. Lastly, no study has investigated the chronic effect of FES on gait in healthy individuals. This deserves special attention in order to understand the effects that this technology could have on the physiological and biomechanical aspects of walking.

## 5. Conclusions

This review provides a comprehensive overview of the effects of functional electrical stimulation on gait characteristics in healthy individuals. The findings unequivocally demonstrate the effectiveness of FES in altering gait kinetics, in particular ground reaction forces and their impulses, the center of pressure displacement, and the knee joint reaction force. Moreover, applying FES to agonist muscles results in amplified joint kinematics, whereas the stimulation of antagonist muscles leads to reduced ranges of motion. As a result of its effects on gait kinematics and kinetics, FES is able to modify spatiotemporal parameters, such as gait speed, step cadence, and stance duration.

However, it is important to be aware that the observed effects exhibit variations based on a multitude of factors, including the demographic composition of the studied population (e.g., older or younger individuals), the specific muscles stimulated, the timing of stimulation, and essential parameters such as frequency, intensity, and pulse width. As a result, the use of FES remains complex and necessitates further investigations to understand the impact of these factors on muscle contributions during gait patterns, force production, fatigue onset in stimulated muscles, and biomechanical gait characteristics in healthy individuals. Furthermore, it is crucial to deepen our knowledge through rigorous and extensive research in order to exploit its full potential as a solution for improving gait performance in healthy people.

## Figures and Tables

**Figure 1 sensors-23-08684-f001:**
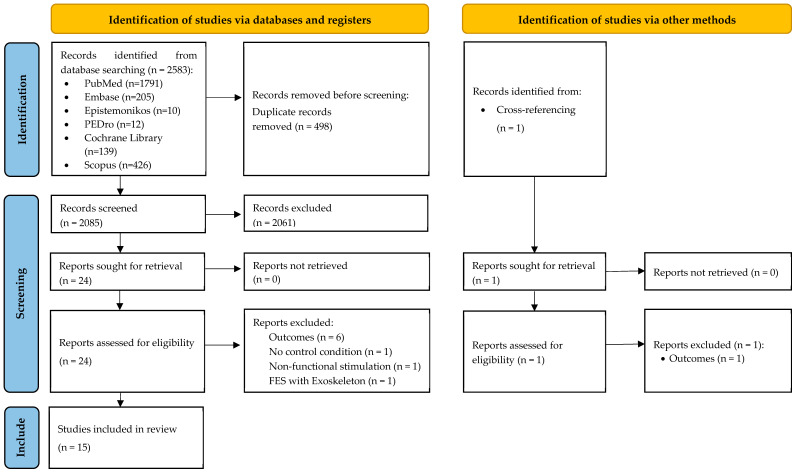
PRISMA 2020 flow diagram.

**Table 1 sensors-23-08684-t001:** Summary of included studies, including first author, demographics (sex and age) of healthy participants, study objectives and protocols, FES parameters, outcome measures, and key findings. (Ordered by year of publication).

First Author (Year and Reference Number)	Sex (Mean Age) of Healthy Participants	Objectives	FES Parameters (Stimulated Muscle; Wave Type; Frequency; Intensity; Stimulation Timing and Trigger Device)	Study Protocol	Outcome Measures	Key Findings
Stewart et al. (2007) [14]	5M (38 years).	To investigate the dynamic function of the calf muscles during normal gait by using FES.	*Muscles*: LG and SOL. *Wave type*: asymmetric biphasic wave. *Pulse width*: adapted to the subjects to make a strong muscle contraction. *Frequency*: 40 Hz. *Intensity*: below the discomfort threshold (≤70 mA). *Stimulation timing:* (1) initial contact to foot-flat, (2) foot-flat to toe-off, (3) heel-off to toe-off. *Trigger device*: foot switch.	Walking trials were performed with 3 different stimulation patterns for each muscle (LG and SOL), giving 6 stimulation conditions. A total of 6 trials were collected for the stimulated and unstimulated conditions.	Knee and ankle angles in the sagittal plane.	Stimulation of LG during stance phase increased the knee flexion angle and the ankle dorsiflexion angle, whereas stimulation of SOL increased knee extension angle and ankle plantar flexion angle. These results vary from subject to subject.
Hernandez et al. (2010) [21]	7 adults (mean age: 30 years).	To evaluate the rectus femoris function during walking by synchronizing electrical stimulation to specific points of the gait cycle.	*Stimulated muscle*: RF of right leg. *Wave type*: not specified. *Pulse width*: 300 μs. *Frequency*: 33 Hz. *Intensity*: below the pain threshold (subjective value of 2 in a 10-point pain scale). *Stimulation timing:* 50% (pre-swing) or 60% (early-swing) of the gait cycle for 90 ms. *Trigger device*: vertical GRF.	Participants performed 90 s walking trials on a split-belt instrumented treadmill while their right RF was stimulated during the pre- or early-swing phases of randomly selected strides.	Hip and knee angles in the sagittal plane.	RF stimulation during pre-swing reduced the knee flexion angle peak in every subject and the hip flexion angle peak in 4/7 subjects. RF stimulation during early swing reduced the knee flexion angle peak in 3/7 subjects and the hip flexion angle peak in 4/7 subjects.
Francis et al. (2013) [22]	20 young adults (mean age: 24 years).	To investigate the relative influence of the gastrocnemius and soleus on support, propulsion, and CoP trajectory in distinct phases of gait.	*Stimulated muscles*: MG and the distal–lateral quadrant of SOL. *Wave type*: not specified. *Pulse width*: 300 μs. *Frequency*: 33 Hz. *Intensity*: <50 mA. *Stimulation timing:* 20% (mid-stance) or 30% (terminal stance) of the gait cycle for 90 ms. *Trigger device*: GRF signal.	Participants performed eight 90 s walking trials at their preferred walking speed. The FES program randomly delivered stimulation to the MG or SOL at 20% or 30% of the gait cycle, with 5–10 strides between stimulation pulse trains.	GRF and CoP.	Stimulation of MG at 20% of gait cycle led to an anterior CoP shift, whereas it led to an increase in the push-off at 30% of gait cycle. Stimulation of SOL decreased the anteroposterior force at both timings, whereas it led to an anterior CoP shift and an increased vertical ground reaction force at 20% of gait cycle.
Lenhart et al. (2014) [13]	20 young adults (7M and 13F, mean age: 24 years).	To evaluate the effect of electrically stimulating SOL and MG at specific portions of the stance phase of gait on lower limb kinematics.	*Muscles*: SOL and MG. *Wave type*: biphasic wave. *Pulse width*: 250 μs. *Frequency*: 40 Hz. *Intensity*: minimum motor threshold (value < 50 mA). *Stimulation timing:* 20% (mid-stance) or 30% (terminal stance) of the gait cycle for 90 ms. *Trigger device*: vertical GRF.	For each gait trial, muscle (MG or SOL) and stimulation timing (20% or 30% of the gait cycle) were randomized. Trials were 90 s in duration and included approximately 10 stimulations per trial.	Lower limb joint angles in the sagittal plane.	MG stimulation during mid-stance induced a greater hip and knee flexion angle 150 ms post-stimulation. Ankle dorsiflexion angle and posterior pelvic tilt were also induced at 200 ms after stimulation onset. In contrast, SOL stimulation during mid-stance induced ankle plantar flexion angle and knee extension angle.
Talis et al. (2015) [23]	16 adults (13M and 3F, mean age: 35 years).	To study the effect of the FES of leg muscles on kinematics of healthy subjects during treadmill locomotion.	*Stimulated muscles*: BF, MG, TA, and quadriceps of both legs. *Wave type*: rectangular pulse. *Pulse width*: 0 to 250 μs. *Frequency*: 65 Hz. *Intensity*: 65 mA. *Stimulation timing:* timing of the activation sequence of various muscles during normal gait. *Trigger device*: right knee goniometer signal.	An experimental group (n = 8) and a control group (n = 8) walked for 40 min on a treadmill. After 10 min without stimulation, FES was applied for 30 min in the experimental group and finally switched off for the last 10 min. Control group walked without FES.	Spatiotemporal gait parameters, trunk oscillations and limb elevation angles in sagittal plane.	FES increased the stance duration during gait. No effect on limb elevation angles in sagittal plane, gait speed, step length, or step width and frequency.
Rane and Bull (2016) [24]	15 young adults (13M and 2F, mean age: 25 years).	To study the effects of stimulating GLM on the medial knee JRF during walking.	*Stimulated muscle:* GLM. *Wave type*: asymmetrical biphasic current waveforms. *Pulse width*: 400 μs. *Frequency*: 45 Hz. *Intensity*: the intensity producing an abduction angle of 30–45° of the right leg while being tolerable. *Stimulation timing:* start before the right foot strike such that stimulation was maximal throughout the stance phase. *Trigger device*: not specified.	Participants performed between 10 and 15 overground walking trials at their preferred speed without and then with FES.	Medial knee JRF, GLM force, GRF, and lower limb kinematics.	Stimulating GLM during stance reduced the medial knee JRF impulse in the mid and terminal stance, increased GLM force impulse, decreased pelvic drop in the frontal plane toward the swing leg, decreased both the mediolateral and vertical GRF impulses, and increased the anteroposterior GRF impulse during stance phase compared to normal gait.
Meng et al. (2017) [25]	5M and 2F young adults (29 years).	To test a new multichannel FES gait system based on a purely reflexive mechanism that is aimed at assisting gait locomotion.	*Muscles*: RF, BF, LG, and TA. *Wave type*: not specified. *Pulse width*: 350 μs. *Frequency*: 40 Hz. *Intensity*: superior to the motor threshold and below the pain threshold. *Stimulation timing:* during swing (TA, BF), at the terminal swing (TA, BF, RF), the pre-swing (LG), and the loading response (RF, LG). *Trigger device*: force-sensitive resistors and inertial measurement units.	The participants performed walking trials on a treadmill in two conditions: (1) 3 min at preferential speed and (2) 1 min with stimulation applied on eight muscles, at the same speed as in the first condition.	Hip, knee, and ankle angles in the sagittal plane.	Five participants obtained a higher peak of ankle plantar flexion angle in the pre-swing phase and a higher peak of ankle dorsiflexion angle in the swing phase. Knee and hip extension were reduced in the stance phase, whereas flexion angles were increased during swing phase.
Azmi et al. (2018) [26]	12 young adults (5M and 7F, mean age: 26 years).	To investigate the effect of stimulating the biceps femoris in stance phase on the internal rotation torque and the anterior tibial shear force during gait.	*Stimulated muscle*: BF long head. *Wave type*: not specified. *Pulse width*: not specified. *Frequency*: 40 Hz. *Intensity*: below the pain threshold; had to generate a knee flexion angle. *Stimulation pattern*: start with 1 s ramp up, 4 s with maximum current, and 1 s ramp down. *Stimulation timing:* heel-strike to toe-off. *Trigger device*: hand switch.	Subjects performed 6 walking trials without stimulation and 6 with stimulation at their self-preferred speed. Stimulation current was at its maximum value from when the heel of the right foot strikes the force plate until toe-off.	Knee joint torque, anterior shear force, knee contact force, patella tendon force, and gait speed.	Stimulation of BF in stance phase reduced the gait speed, the peak value of the tibial internal rotation torque, and the anterior shear force at the knee. In contrast, it increased the peak of lateral knee compressive force and the peak of patella tendon force.
Chen et al. (2018) [27]	9 young adults (5M and 4F, mean age 23) and 10 post-stoke adults.	To compare two methods of triggering FES for drop foot correction during walking.	*Stimulated muscle:* TA. *Wave type*: rectangular pulse. *Pulse width*: 400 μs. *Frequency*: 40 Hz. *Intensity*: intensity when the subjects achieved a neutral ankle angle (0°) in a seated position with the foot hanging freely in a plantar-flexed position. *Stimulation timing:* heel-off to heel strike. *Trigger device*: foot switch.	Healthy controls walked on a treadmill at 4 speeds (0.3, 0.6, 0.9, and 1.2 m/s) under 3 stimulation conditions: (1) FES triggered by the heel-off event (HOS), (2) FES triggered by a speed-adaptive algorithm (SAS), and (3) without FES (NS).	Peak of the knee flexion angle, maximum dorsiflexion angle during the swing phase, and ankle angle at the toe-off event.	Higher peak of dorsiflexion angle during swing phase and a decrease in plantar flexion angle in SAS condition compared with NS condition. Peak knee flexion angle in the NS condition was similar to that in the SAS condition at most speeds.
Okamura et al. (2018) [28]	20M young adults (21 years).	To examine the effect of reinforcing the plantar intrinsic foot muscles (PIFMs) via electrical stimulation on foot dynamics during gait.	*Muscle*: abductor hallucis of the right leg. *Wave type*: not specified. *Pulse width*: 250 μs. *Frequency*: 20 Hz. *Intensity*: below the pain threshold. *Stimulation timing:* PIFMs were stimulated from mid-stance to pre-swing. *Trigger device*: hand switch.	Two groups performed 5 walking trials at their self-selected preferred speed on an 8 m walkway. Afterward, 5 trials were conducted again with FES only in the experimental group.	Stance duration, foot kinematics, ankle moments, and GRF.	FES slowed the deformation of the medial longitudinal arch, decreased forefoot abduction, and reduced the second peak of the vertical ground reaction force. No effect on gait speed, stance duration, forefoot eversion, ankle dorsiflexion, or anteroposterior and mediolateral GRF.
Ding et al. (2019) [15]	13 young adults (5M and 8F, mean age: 26 years).	To quantify the effect of stimulating biceps femoris during the stance phase of gait and validating in a musculoskeletal model.	*Stimulated muscle*: BF long head. *Wave type*: not specified. *Pulse width*: 120 μs. *Frequency*: 40 Hz. *Intensity*: 40 mA, 60 mA, and 80 mA (each intensity had to generate a knee flexion angle). *Stimulation timing:* delivered at the early stance on the muscle activation duration. *Trigger device*: manual (hand switch).	Participants performed 6 walking trials without stimulation and 6 walking trials per intensity of stimulation (3 intensities) at their self-selected preferred speed on a 6 m walkway.	Gait speed.	Stimulation of BF during stance phase did not affect the gait speed. GMAX EMG peak and impulse during stance phase increased with stimulation intensity of BFLH.
Thorp et Adamczyk (2020) [29]	8F young (College-aged).	To examine the effects of electrical stimulation of gastrocnemius at various phases of the gait cycle on treadmill and overground walking.	*Muscle*: right medial gastrocnemius (MG). *Wave type*: biphasic wave. *Pulse width*: 350 μs. *Frequency*: 40 Hz. *Intensity*: minimum motor threshold (Min), maximum tolerable intensity (Max), and 2/3 between Min and Max. *Delay*: 0–1 s. *Stimulation timing:* 8 distinct subphases within stance phase (0–49% of the gait cycle in 7% increments and 49–60%) and 4 subphases within swing phase (60–100% in 10% increments) for 100 ms. *Trigger device*: inertial measurement unit.	Participants preformed four trials of treadmill gait for two minutes at their preferred speed. A 1 min break was established between trials. In parallel, four overground walking trials were performed. Each stimulation pulse train was separated by a random integer (from 4 to 6) of normal strides.	Stride duration.	Depending on the stimulation timing, stride duration was influenced by the stimulation of MG. The stride period was shorter when stimulation was applied around the push-off phase and was longer when stimulation was applied around foot contact.
Dong et al. (2022) [30]	10 young adults (mean age: 25 years).	To validate a FES walking assistance system with an adaptive control method of the stimulation based on temporal gait parameters and sagittal shank angle.	*Stimulated muscles*: RF, BF, LG, and TA of both legs. *Wave type*: not specified. *Pulse width*: 250 to 500 μs. *Frequency*: not specified. *Intensity*: adaptative. *Stimulation timing:* during swing (TA, BF), at the terminal swing (TA, BF, RF), at the pre-swing (LG), and at the loading response (RF, LG). *Trigger device*: force-sensitive resistors and inertial measurement unit.	A total of 3 conditions of walking on treadmill: without FES (NFC), with reflexive FES controller (RFC), and with adaptative and reflexive FES controller (ARFC). The walking speed increased from 1.0 to 2.0 km/h and then decreased to 1.0 km/h, with 0.2-km/h steps.	Joint kinematics of the hip, knee, and ankle in the sagittal plane.	Combined stimulation of various muscles increased the ROM angle of the ankle, knee, and hip. ARFC had a greater effect than RFC on all kinematics parameters at different walking speeds.
Gottlieb et al. (2022) [31]	24 adults (13M and 11F, mean age 30 years) and 24 adults (17M and 7F, mean age 30 years) with chronic ankle instability (CAI).	To study the effects of a single gait training session with peroneal FES on ankle kinematics and peroneal activity in individuals with and without CAI.	*Stimulated muscle:* below the head of the fibula and over the peroneus longus belly. *Wave type*: biphasic symmetrical pulse. *Pulse width*: 200 μs. *Frequency*: 35 Hz. *Intensity*: above the motor threshold and below the discomfort threshold (between 33 mA and 40 Hz). *Stimulation timing:* was delivered between 0% and 80% of the stance phase. *Trigger device*: foot switch.	Participants walked for 10 min with FES on a treadmill at a pace 20% faster than their preferred walking speed.	Ankle kinematics and peroneal activity (EMG).	After a single gait training session with FES, healthy controls had significantly more ankle eversion angle at early and late stance than before the intervention in this phase, without a change in the peroneal muscle activity.
Park et al. (2022) [16]	10M and 19F old adults (75 years).	To examine the immediate effects of wearable EMG-controlled FES on the lower limb muscle morphology, balance, and gait in older adults.	*Muscles*: RF, BF, TA, and MG. *Wave type*: rectangular biphasic wave. *Pulse width*: 250 μs. *Frequency*: 40 Hz. *Intensity*: below the pain threshold (range: 10–40 mA). *Pulse duration:* not specified. *Stimulation timing:* delivered at different moments of stance and swing phase, without specifying the events. *Trigger device*: EMGs.	After a familiarization phase with the EMG-controlled FES (5 to 10 min), walking trials were carried out with and without FES (six trials in total) in a randomized order. Participants walked at a self-selected speed on a walkway approximately 9 m long and on a 5 m GAITRite mat with 2 m acceleration and deceleration periods.	Spatiotemporal gait parameters	FES led to an increase in gait speed and cadence. No effect on stride length, step length, and step width.

**Muscle abbreviations—GLM**: Gluteus Medius; **RF**: Rectus Femoris; **BF**: Biceps Femoris; **BFLH**: Biceps Femoris Long Head; **LG**: Lateral Gastrocnemius; **MG**: Medial Gastrocnemius; **SOL**: Soleus; **TA**: Tibialis anterior; **PIFM**: Plantar Intrinsic Foot Muscle. **Other abbreviations**—**M**: Males; **F**: Females; **CoP**: Centre of Pressure; **GRF**: Ground Reaction Force; **JRF**: Joint Reaction Force.

**Table 2 sensors-23-08684-t002:** Methodological assessment of studies classified by year of publication, according to the PEDro scale.

Authors	1	2	3	4	5	6	7	8	9	10	11	Score	Reliability (%)
Stewart et al., 2007 [14]	N	Y	Y	Y	Y	N	N	Y	Y	N	N	6/11	54%
Hernandez et al., 2010 [21]	N	N	N	Y	N	N	N	Y	Y	Y	Y	5/11	45%
Francis et al., 2013 [22]	N	Y	Y	Y	Y	N	N	Y	Y	Y	N	7/11	63%
Lenhart et al., 2014 [13]	Y	Y	Y	Y	Y	N	N	Y	Y	Y	Y	9/11	81%
Talis et al., 2015 [23]	N	N	N	Y	N	N	N	Y	Y	Y	Y	5/11	45%
Rane and Bull, 2016 [24]	N	N	N	Y	N	N	N	Y	Y	Y	Y	5/11	45%
Meng et al., 2017 [25]	N	N	N	Y	Y	N	N	Y	Y	Y	Y	6/11	54%
Azmi et al., 2018 [26]	N	N	N	Y	N	Y	N	Y	Y	Y	Y	6/11	54%
Chen et al., 2018 [27]	N	Y	Y	N	Y	N	N	Y	Y	Y	Y	7/11	63%
Okamura et al., 2018 [28]	Y	Y	Y	Y	Y	N	N	Y	Y	Y	Y	9/11	81%
Ding et al., 2019 [15]	N	N	N	Y	N	N	N	Y	Y	Y	Y	5/11	45%
Thorp and Adamczyk, 2020 [29]	N	Y	Y	Y	Y	N	N	Y	Y	Y	Y	8/11	72%
Dong et al., 2022 [30]	N	N	N	Y	N	N	N	Y	Y	Y	Y	5/11	45%
Gottlieb et al., 2022 [31]	Y	Y	Y	N	Y	N	N	Y	Y	Y	Y	8/11	72%
Park et al., 2022 [16]	Y	Y	Y	Y	Y	N	N	Y	Y	Y	Y	9/11	81%

**Notes:** 1—The eligibility criteria has been clarified; 2—Subjects were randomly assigned to the following groups; 3—The distribution respected a secret assignment; 4—The groups were similar at baseline for the most important prognostic indicators; 5—All subjects were “blinded”; 6—All researchers who administered the treatment were "blinded”; 7—All reviewers were “blinded” on at least one of the primary outcomes; 8—Measures for at least one of the primary endpoints were obtained for more than 85% of subjects initially assigned to the groups; 9—All subjects for whom results were available received the treatment or control intervention according to their allocation or, when this was not the case, data for at least one of the primary endpoints were analyzed on an “intention-to-treat”; 10—Results of intergroup statistical comparisons are reported for at least one of the primary outcomes; 11—For at least one of the primary endpoints, the study reports both the effect estimate and the variability estimate. Score: This section is related to the positive responses obtained for the 11 items. Reliability: This section reports the reliability percentage corresponding to the PEDro score obtained for each evaluated study.

**Table 3 sensors-23-08684-t003:**
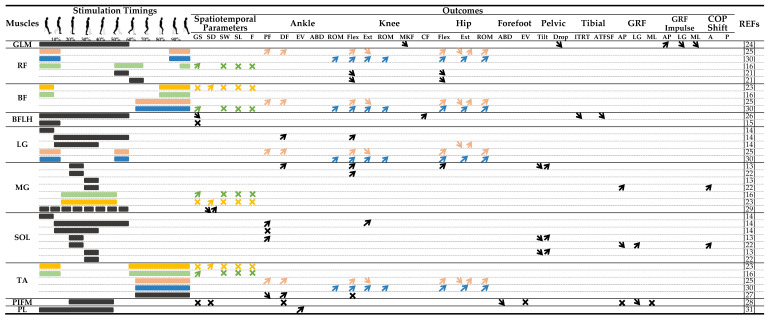
Outcomes according to the stimulated muscles with their stimulation timing. Note that since up to four muscles can be stimulated during the gait cycle [13,14,15,16,21,22,23,24,25,26,27,28,29,30,31], the stimulation timing and combined effects are repeated for all stimulated muscles with the same color code. When only one muscle was stimulated, the timing and effects are shown in black.

**Muscles abbreviations**—GLM: Gluteus Medius; RF: Rectus Femoris; BF: Biceps Femoris; BFLH: Biceps Femoris Long Head; LG: Lateral Gastrocnemius; MG: Medial Gastrocnemius; SOL: Soleus; TA: Tibialis anterior; PIFM: Plantar Intrinsic Foot Muscle; PL: Peroneus Longus; **Outcome abbreviations**—ABD: Abduction (frontal plane); EV: Eversion (transversal plane); PF: Plantarflexion (sagittal plane); DF: Dorsiflexion (sagittal plane); ROM: Range of motion (sagittal plane); MKF: Medial knee force; CF: Compressive force; AP: Anteroposterior component; LG: Longitudinal component; ML: Mediolateral component; A: Anterior; P: Posterior; ITRT: Internal Tibial Rotation Torque; ATFSF: Anterior Tibiofemoral Shear Force; GS: Gait speed; SD: Stance duration; SL: Step length; F: Frequency. **Other abbreviations**—**⭧**: Increase; **⭨**: Decrease; **⭨⭧**: Opposite effects achieved at different moments of the gait cycle; **X**: No effect.

## Data Availability

Not applicable.

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
