# Peer review of "Effects of Functional Electrical Stimulation on Gait Characteristics in Healthy Individuals: A Systematic Review"

_sensors, 2023, doi:10.3390/s23218684_

Round 1

Reviewer 1 Report

The study of Auout and colleagues aims to review the current state of the art in terms of effects of electrical stimulation in gait. The Literature review was well conducted through the PRISMA protocol and Authors pointed out the current trend in healthy individual. I have only one concerns related to the fact that the techniques are developed also for counteract negative effects that may occur in neurodenerative diseases. In this field recent parameters and techniques were pointed out to characterize gait of people affected by pathologies. This surely help in highlighting the importance and the needs behind a review the Authors proposed. For this reason I suggest to at least  mention and discuss also the following studies:

-  "Identification of neurodegenerative diseases from gait rhythm through time domain and time-dependent spectral descriptors." IEEE Journal of Biomedical and Health Informatics 26.12 (2022): 5974-5982.

- "Human gait analysis in neurodegenerative diseases: A review." IEEE Journal of Biomedical and Health Informatics 26.1 (2021): 229-242.

Author Response

We greatly thank the reviewer for the very precise and relevant comments on the manuscript. They have served to substantially improve the quality of the manuscript. Please find below the point-by-point reply to each of the remarks raised.

Reviewer #1: The study of Aout and colleagues aims to review the current state of the art in terms of effects of electrical stimulation in gait. The Literature review was well conducted through the PRISMA protocol and Authors pointed out the current trend in healthy individual. I have only one concerns related to the fact that the techniques are developed also for counteract negative effects that may occur in neurodenerative diseases. In this field recent parameters and techniques were pointed out to characterize gait of people affected by pathologies. This surely help in highlighting the importance and the needs behind a review the Authors proposed. For this reason, I suggest to at least mention and discuss also the following studies:

 -  "Identification of neurodegenerative diseases from gait rhythm through time domain and time-dependent spectral descriptors." IEEE Journal of Biomedical and Health Informatics 26.12 (2022): 5974-5982.

 - "Human gait analysis in neurodegenerative diseases: A review." IEEE Journal of Biomedical and Health Informatics 26.1 (2021): 229-242.

ReplyThank you for this relevant suggestion. These studies have been incorporated into the discussion and reference sections to reinforce this point. You can find these changes in pages 20, lines 597-602.

“Concurrently, recent studies involving individuals with neurodegenerative pathologies have revealed recurrent temporal and kinematic alterations during gait compared to healthy subjects (Cicirelli et al. 2022; Mengarelli et al. 2022). In this context, a deeper understanding of the effects of FES on a healthy population could serve as a robust foundation for developing biotechnological solutions aimed at mitigating the adverse effects observed during the gait of individuals with neurodegenerative diseases.”

References

Cicirelli, G.; Impedovo, D.; Dentamaro, V.; Marani, R.; Pirlo, G.; D’Orazio, T.R. Human Gait Analysis in Neurodegenerative Diseases: A Review. IEEE J. Biomed. Health Inform. 2022, 26, 229–242, doi:10.1109/JBHI.2021.3092875.

Mengarelli, A.; Tigrini, A.; Fioretti, S.; Verdini, F. Identification of Neurodegenerative Diseases From Gait Rhythm Through Time Domain and Time-Dependent Spectral Descriptors. IEEE J. Biomed. Health Inform. 2022, 26, 5974–5982, doi:10.1109/JBHI.2022.3205058.

Reviewer 2 Report

The general question that must be raised is "Why this article has been written at all. The answer I feel tempted to give is: a well-founded attack against writing papers without insufficient content. The ratio of 2585 papers which look at a first glance promising to  15 really worthful articles is really  impressive!  

Headline and elsewhere:

FES is also used as  a the short form of “Finite Element System” – at least since Witzel & Preuschoft,  2005.  Using the same abbreviation for different things leads to confusion. Why “functional” in  the present case?

The entire text is written in a clear and understandable language, but 3rd  narrows the view on informations from the literature.  I personally do not see a difference between  “spatiotemporal” and “kinematics”?  Here, the latter is restricted to movements of the joints.: ankle, knee, hip, pelvis.

Methods:

The authors base their study on a wealth of literature: The number of  2585 records is mentioned in the “Results” Of this great number, not more than 15  were finally included in the study. They are listed in Tables 1 and 2. This strongly limited number comprised, however, 215 Individuals ((3.3., participants)).

The ways of stimulating muscles are detailedly described.

The effects of stimulation are shown in Table 3. My own point of view, developed on the basis of mechanical theory,  these effects are exactly what could be expected theoretically without any experiments!

The Discussion expands over many details under consideration. It ends with “limitations” of the study.

Author Response

We greatly thank the reviewer for his/her thoughtful review of our manuscript and his/her positive attitude on our work. Please find below a detailed reply to each of the points raised by the reviewer.

Reviewer #2: The general question that must be raised is "Why this article has been written at all. The answer I feel tempted to give is: a well-founded attack against writing papers without insufficient content. The ratio of 2585 papers which look at a first glance promising to 15 really worthful articles is really impressive! 

 Reply:  Thank you very much for this valuable comment. Indeed, you have accurately captured the essence of our rationale. Within the scientific literature, several studies have explored the effects of electrical muscle stimulation during walking, primarily focusing on pathological populations. Nevertheless, there is a significant gap in the scientific literature when it comes to investigating the effects of functional electrical stimulation on gait parameters in healthy individuals. Thus, the primary objective of our article was to conduct a systematic analysis of existing research in this domain to elucidate the outcomes and effects of this approach in healthy individuals.

 Headline and elsewhere

  • FES is also used as the short form of “Finite Element System” – at least since Witzel & Preuschoft, 2005. Using the same abbreviation for different things leads to confusion. Why “functional” in the present case ?

Reply: Thank you for your relevant comment. FES is the acronym classically used in research and clinical practice for “Functional Electrical Stimulation”. The term “functional” is used because FES primarily aims to trigger muscle contractions to produce and/or enhance functional movements (not only for muscle conditioning), like walking (Melo et al. 2015; Rane and Bull. 2016) and grasping (Popovic et al. 2001).

  • The entire text is written in a clear and understandable language, but 3rd narrows the view on informations from the literature. I personally do not see a difference between “spatiotemporal” and “kinematics”? Here, the latter is restricted to movements of the joints: ankle, knee, hip, pelvis.

 Reply: The reviewer is right. Gait kinematics includes spatiotemporal parameters. Nevertheless, in the field of gait analysis, spatiotemporal parameters are often distinguished from other parameters describing body movement (segment and joint kinematics), as these parameters provide complementary information about the individual's gait (e.g., Mao et al., 2012). In addition of this reason, we have adopted this commonly used terminology in the present review in order to facilitate the description of the results obtained in the literature.

Methods

The authors base their study on a wealth of literature: The number of 2585 records is mentioned in the “Results” Of this great number, not more than 15 were finally included in the study. They are listed in Tables 1 and 2. This strongly limited number comprised, however, 215 Individuals ((3.3., participants)).

 The ways of stimulating muscles are detailedly described.

 The effects of stimulation are shown in Table 3. My own point of view, developed on the basis of mechanical theory, these effects are exactly what could be expected theoretically without any experiments!

 Reply: The reviewer is right. By providing an overview of the results obtained in the literature on the effects of FES on walking, the present review helps to provide evidence for hypotheses that are often formulated theoretically.

 The Discussion expands over many details under consideration. It ends with “limitations” of the study

 Reply: Thank you for your feedback concerning our work.

 References

 Melo, P.L.; Silva, M.T.; Martins, J.M.; Newman, D.J. Technical Developments of Functional Electrical Stimulation to Correct Drop Foot: Sensing, Actuation and Control Strategies. Clin Biomech (Bristol, Avon) 2015, 30, 101–113, doi:10.1016/j.clinbiomech.2014.11.007.

Rane, L.; Bull, A.M.J. Functional Electrical Stimulation of Gluteus Medius Reduces the Medial Joint Reaction Force of the Knee during Level Walking. Arthritis Res Ther 2016, 18, 255, doi:10.1186/s13075-016-1155-2.

Popovic, M.R.; Curt, A.; Keller, T.; Dietz, V. Functional Electrical Stimulation for Grasping and Walking: Indications and Limitations. Spinal Cord 2001, 39, 403–412, doi:10.1038/sj.sc.3101191.

Mao, Y.R.; Zhao, J.L.; Bian, M.J.; Lo, W.L.A.; Leng, Y.; Bian, R.H.; Huang, D.F. Spatiotemporal, Kinematic and Kinetic Assessment of the Effects of a Foot Drop Stimulator for Home-Based Rehabilitation of Patients with Chronic Stroke: A Randomized Clinical Trial. J NeuroEngineering Rehabil 2022, 19, 56, doi:10.1186/s12984-022-01036-0.